# Using Fetal Fibronectin Test to Reduce Hospital Admissions with Diagnosis of Preterm Labor: An Economic Evaluation Study

**DOI:** 10.3390/jpm13060894

**Published:** 2023-05-25

**Authors:** Bedayah Amro, Iman Alhalabi, Anila George, Hanan Haroun, Amar Hassan Khamis, Nadia Al Sawalhi

**Affiliations:** 1Department of Obstetrics and Gynecology, Latifa Women and Children Hospital, Dubai P.O. Box 9115, United Arab Emirates; 2Quality and Corporate Development Office, Latifa Women and Children Hospital, Dubai P.O. Box 9115, United Arab Emirates; 3Laboratory Dept, Latifa Women and Children Hospital, Dubai P.O. Box 9115, United Arab Emirates; 4Department of Obstetrics and Gynaecology, College of Medicine, Mohammed Bin Rashid University of Medicine and Health Sciences, Building 14, Dubai Healthcare City, Dubai P.O. Box 505055, United Arab Emirates

**Keywords:** preterm labor, fetal fibronectin, cost, hospital admission, Dubai

## Abstract

**Background**: Preterm labor and delivery remain a major problem in obstetrics accounting for perinatal morbidity and mortality. The challenge is to identify those with true preterm labor to avoid unnecessary hospital admissions. The fetal fibronectin (FFN) test is a strong predictor of preterm birth and can help identify women with true preterm labor. However, its cost-effectiveness as a strategy for triaging women with threatened preterm labor is still debatable. **Objective:** To evaluate the effect of FFN test implementation on hospital resources by reducing the admission rate of threatened preterm labor in a tertiary hospital, Latifa Hospital, UAE. **Methods:** A retrospective cohort study of singleton pregnancies between 24 and 34 weeks of gestation who attended Latifa Hospital in the period of September 2015–December 2016, complaining of threatened preterm labor after the availability of an FFN test, and a historical cohort study for those who attended with threatened preterm labor before the availability of an FFN test. Data analysis was performed using a Kruskal–Wallis test, Kaplan–Meier, Fischer exact chi-square and cost analysis. The significance was set at *p*-value < 0.05. **Results:** In total, 840 women met the inclusion criteria and were enrolled. The relative risk of FFN for delivery at term was 4.35 times higher among the negative-tested compared to preterm delivery (*p*-value < 0.001). A total of 134 (15.9%) women were unnecessarily admitted (FFN tested negative, delivered at term) which yielded $107,000 in extra costs. After the introduction of an FFN test, a 7% reduction of threatened preterm labor admissions was recorded.

## 1. Introduction

Preterm labor and delivery remain a major problem in obstetrics, accounting for 70% of perinatal mortality and 50% of long-term neurological morbidity [1,2,3,4]. The WHO states that preterm complications are the leading cause of death among children under 5 years of age, responsible for approximately 900,000 deaths in 2019. Three-quarters of these deaths could be prevented with current, cost-effective interventions [5]. The universal definition of preterm birth includes all deliveries between 24 and 36 + 6 weeks. This can be a result of two main clinical subtypes; indicated preterm deliveries undertaken either for maternal or fetal etiologies, found approximately in one-third of all such births. The other two-thirds are classified as spontaneous preterm deliveries that are subdivided into spontaneous preterm labor and preterm pre-labor rupture of the membranes (PPROM) [1,2]. The rate of preterm deliveries varies across the globe (4–16% in 2020) [6] for reasons related to etiology, ethnicity, socioeconomic and cultural factors. Unfortunately, this led to 13.4 million babies born preterm in 2020. A diagnosis of preterm labor previously relied on the patient’s perception of contractions in addition to CTG (cardiotocography), and findings suggestive of contractions (which could be due to dehydration, Braxton hicks contractions, etc.), despite their poor predictive value. However, 75–95% of these patients do not deliver within 7 days, and 40% of such patients will deliver at term [1,7,8]. As a result, many women with preterm labor were managed with tocolytic, antibiotics, corticosteroids and admission to an obstetric unit which led to substantial costs to the hospital, inefficient health resource utilizations and possible adverse side-effects to both the mother and the unborn child [2,9]. The recommendations were towards the risk stratification of women into high-risk or low-risk groups for true preterm labor and delivery in order to target the proper prevention and management action plan. Major risk factors for the high-risk group are: previous preterm births, uterine overdistension or anomalies, current illness or surgery during the pregnancy and other environmental factors such as smoking, low body mass index (BMI) and socioeconomic deprivation. 

The appropriate identification of women at low-risk of immediate preterm delivery could reduce the unnecessary treatment for those women, resulting in cost savings without affecting the health outcome. A variety of methods have been suggested over the last decades for the proper prediction of true preterm labor. Cervical length measurements by transvaginal scanning have been shown to be more accurate than digital measurements for assessing cervical length. In asymptomatic women with a short cervix, the risk of preterm labor and delivery increases dramatically as the length further decreases. However, this needs a proper scanning machine with a transvaginal probe, skilled physician and repetitive serial scanning or reference baseline measurements [7,10]. Other ultrasound markers were described as a tool for prediction such as: fetal membrane thickness, uterine artery pulsatility index during the peak of uterine contractions, placental strain ratio, fetal middle cerebral artery pulsatility index (MCA-PI) and measurements of the central zone of the fetal adrenal gland. They all usually require expertise and their exact predictive values are yet to be studied [11]. In addition, amniotic fluid indicates certain factors such as low glucose, low interleukin-6 (IL-6), high vascular endothelia growth factor (VEGF) and placental growth factor (PGF) and low soluble VEGF resecptor-1 (sFLt-1) [12]. Furthermore, maternal serum factors include calponin 1, alpha fetoprotein, progesterone-induced blocking factor (PIBF) and salivary estriol, while cervical fluid biomarkers are placental alpha macroglobulin-1(PAMG-1), insulin-like growth factor binding protein-1 (IGFBP-1) and fetal fibronectin (FFN). However, none of the screening tests can fulfill the criteria for an ideal screening test, and variations in their predictive values are greatly affected by the population studied, sample size and solitary or combined use with other methods [13].

Fetal fibronectin has been demonstrated to have high negative predictive values (NPV) for preterm delivery [3,14] and can avoid unnecessary admissions and treatments [15,16,17,18,19,20,21]. Fetal fibronectin (FFN) is a glycoprotein produced by the chorion that is believed to have a role in implantation and placental attachment to the uterus. It can be detected by cervical and vaginal secretions prior to 20 weeks of gestation, but the presence of FFN in vaginal secretions after 22 weeks usually indicates that a disruption of the utero-placental interface has occurred. FFN can be detected in cervical and vaginal secretions using a specific monoclonal antibody assay [22,23,24,25]. Technique is important in performing the bedside test. A digital examination of the cervix should not have been performed prior to FFN testing, no lubricating jelly used, no sexual intercourse or vaginal pessary within 24 h and no vaginal bleeding or leaking, as this may result in false positive values. So far, no randomized trial has shown beneficial effects of fibronectin testing for women presenting with threatened preterm labor, despite its high negative predictive value. However, it is most accurate in predicting preterm birth in women with threatened preterm labor without advanced cervical dilatation within 7–10 days after testing [3], and clinicians can use its high negative predictive value to either withhold treatments or optimize their timing. Although the use of fibronectin tests for these women is already implemented in some clinics and even incorporated into guidelines [8], its cost-effectiveness as a triage instrument has not yet been established [26].

The cost effectiveness associated with the introduction of an FFN test, in relation to its relative impact on hospital admissions, length of hospital stay and the clinical management of threatened preterm labor, has never been reported from the UAE. The local hospital protocol for the management of women attending the emergency department with symptoms of threatened preterm labor included the conventional diagnostic method such as physical exam, digital cervical assessment and uterine contractions by CTG, followed accordingly by admission to hospital for observation, hydration and corticosteroid administration for lung maturity and tocolytics administration (Atosiban) accordingly. The ultrasound cervical assessment, though it is considered a cost-effective method, was not used, as it mandates a transvaginal assessment with proper ultrasound skills which may be not universally guaranteed in our hospital set due to the skills variation of the OBGYN residents who usually run the emergency unit, and it may add unnecessary inconvenience or anxiety to both patients and doctors if the measurements were not accurate and needed a repeat by the senior physician. In addition, it may need serial measurements or a baseline assessment to compare with, especially in high-risk groups, which are not guaranteed as a large pool of our patients attended the emergency department without any previous antenatal checkups in our hospital or followed in other hospitals in the private sector without a shared health record. The protocol was amended in September 2015 with the addition of FFN at the diagnostic level (as a simple, effective, non-operator dependent and fast bedside result), in order to triage the women with true preterm labor and hence, avoid unnecessary admission and the over-treatment of those women. The aim of this study is to evaluate the economic impact of using FFN as a triage test to reduce the hospital admissions and the associated interventions for women diagnosed with threatened preterm labor.

## 2. Material and Methods

The study collected data retrospectively between September 2015 and December 2016 from women presenting with possible preterm labor when FFN testing was first implemented. The study was conducted at Latifa Hospital, the largest maternal tertiary referral hospital in Dubai, United Arab Emirates (UAE), and received ethical approval from the ethical committee at the Dubai Health Authority (DHA). 

The inclusion criteria were: singleton pregnancy and a gestational age between 24 and 34 weeks who presented to an emergency department with signs and symptoms of preterm labor (uterine contraction, low back pain, pelvic pressure or low abdominal pressure). Those who had the following were excluded: vaginal leaking or bleeding, cervical dilatation more than 3 cm, cervical cerclage and vaginal interruption within the last 24 h, (e.g., vaginal intercourse or examination, vaginal lubricants or pessaries). The data on the FFN test results were obtained directly from the files of the women. The initial clinical evaluation of threatened preterm consisted of a non-stress test, urine dipstick test and speculum examination for visual evaluation of the cervix and for the collection of the FFN swab.

We used the rapid version of the test, which is a lateral flow, solid-phase immunosorbent assay device that is designed to qualitatively detect FFN in cervicovaginal specimens collected with a specific specimen collection kit. The patients were categorized into four groups: (1) FFN status positive and delivered preterm, (2) FFN status positive, and delivered at term, (3) FFN status negative and delivered preterm and (4) FFN status negative and delivered at term. 

The cost is calculated based on the charges applied to a patient in the initial admission. The cost is inclusive of hospital stay (from the time of admission in view of threatened preterm labor until the time of discharge), routine labs and other diagnostic tests and medications used throughout the hospital stay. 

The routine laboratory tests conducted for this category of patients included CBC, blood group and save, urine routine and culture (MSU), serial blood sugar, HIV, VDRL, HBsAg and a high vaginal swab (HVS). In addition, the obstetric ultrasound was performed once and the CTG is performed at least twice during the stay. The steroid includes a betamethasone injection (12 mg once a day for 2 days), tocolytics include an Atosiban infusion for a total of 24 h and blood sugar with any need for insulin administration as a side effect of corticosteroid administration was monitored.

The cost of hospital stay (per day) was derived from the Dubai Health Authority (DHA) reference costs. The costs of tocolytic and corticosteroids were obtained from the DHA Formulary (2017).

The cost of an obstetric ultrasound examination, CTG and lab tests were taken from the DHA-SAM application (an electronic medical record system). The cost comparison was conducted in terms of high and low costs in which the high costs are inclusive of costs incurred with room rentals, diagnostics and medications during the estimated 3 days stay in the hospital, where as low costs are only inclusive of the cost of an FFN test and a short stay at the emergency department. The cost was calculated in UAE Dirhams and converted into US Dollars. The CHEERS 2022 guidelines were applied as it is a health economic evaluation study [27].

## 3. Statistical Analysis

The data were analyzed using IBM SPSS version 23, chigoe. Categorical variables were described by using frequency and proportion, while continuous variables were described by the mean and standard deviation. The continuous data were tested for normality by using the Kolmogorov–Smirnov test. Dichotomous variables were compared by using the Fischer exact test; the continuous data in the four groups was compared by using the Kruskal–Wallis test. Kaplan–Meier curves were generated to compare the number of days from sampling to the delivery in the four groups; the comparison of curve was conducted by a log-rank test. A *p*-value of < 0.05 was considered significant for all the tests.

## 4. Results

A total of 958 patients were tested for fetal fibronectin (FFN) status retrospectively from Latifa Hospital between September 2015 and December 2016. Out of these patients, *n* = 65 (6.7%) were twins, *n* = 7 (0.7%) were triplets and *n* = 885 (92.55) were singletons. Among the 885 singletons, 7 were excluded due to a cervical dilation of more than 3 cm. The data were completed for the 840 patients enrolled in this study. Fetal fibronectin was positive among 99 (11.8%) and negative among 741 (88.2%) patients. Of the patients who tested positive, there were 42 (42.4%) preterm deliveries, while among the negative fetal fibronectin patients about 586 (69.7%) had at-term delivery (*p*-value = 0.001), Table 1. The sensitivity of fetal fibronectin over the whole period from the day of the test to delivery was 21.3% and specificity was 91.1%, while the positive predictive value (PPV) was 42.4%, and the negative predictive value (NPV) was 79.1%. On the other hand, those values were changing as the weeks passed from the date of when the FFN test was collected to the delivery date. On the first week, it was 52% and 100% for sensitivity and specificity, respectively, while dropping to 41.5% and 89.7% at the fourth week. The relative risk (RR) of a fetal fibronectin test for delivery at term was 4.35 times more among the negative tested compared with the preterm delivery group (*p*-value < 0.001). When comparing the four subgroups, the average gestation age at the time of FFN collection was around 30 weeks for all and the parity was higher for the preterm group, whether a positive or negative FFN (*p*-value = 0.001). The number of days at hospital (when the woman is admitted with threatened preterm labor until discharge without delivery) was significantly lower for the negative FFN group (*p*-value = 0.004), Table 2. Having a negative FFN significantly affected the physician’s decision for discharge rather than admission, as well as corticosteroid administration (*p*-value = 0.012, 0.004). However, the opposite happened when the result was positive, though not statistically significant (*p*-value = 0.30, 0.13). Unfortunately, 26% of women who were FFN positive received insulin due to impaired blood sugar secondary to corticosteroid administration unnecessarily as they eventually delivered at-term, compared to only 9% if FFN was negative, Table 3. Eight percent (8%) of women who tested FFN negative delivered preterm, though for 50% of them it was more than 5 weeks from the FFN test date, Figure 1. With survival analysis, preterm delivery groups had the lowest survival curve compared to the at-term delivery group, Figure 2.

## 5. Discussion

Women with threatened preterm labor who attended our emergency department form around one-third of emergency attendants. The unnecessary admissions and treatments contribute largely to skyrocketing health care costs. Testing for FFN in cervicovaginal fluid has been shown to be very helpful in predicting true preterm labor [10,28,29]. 

Our results showed high specificity values for FFN, and this was not affected generally with the time from the test collection date. Sensitivity on the other hand showed decreasing values with each week passing from that day, matching the previous reports [28,29,30,31]. Negative and positive predictive values of testing were around 80% and 42%, respectively. Using FFN at our hospital, at the level of assessing women presenting with threatened preterm labor, it was found to be simple, does not need specific skills to be performed, provides a fast bedside result that does not nictitate a previous baseline value to compare with and has a high negative predictive value. This applied well to our hospital structure where the OBGYN residents can use it confidently without the need for supervision or repetition compared to a cervical assessment by a transvaginal ultrasound. This is more evident in cases where there were no cervical length decrements.

One hundred and thirty four women (18%) who had a negative test were admitted for observation and possible interventions, only to be eventually discharged from the hospital undelivered and later had a full term delivery; this led to the pure wasting of unnecessary costs which could have been simply avoided if they were not admitted. The detailed cost for all possible interventions for any patient admitted with suspected preterm labor shown at Table 4.

Out of 134 women, 33 (24.6%) received a steroid injection with a cost of USD 970.2. As a consequence, 12 patients (36.4%) developed high blood sugar that needed insulin correction doses which incurred a cost of USD 143.7. In addition, 7 patients (5.2%) received Atosiban which raised the cost to USD 6022.3. The 134 patients stayed at the hospital for 373 days overall which incurred an overall cost of USD 50,776.4 for room rentals alone.

As a result, the hospital incurred an all-inclusive unjustified cost of around USD 107,000 by unnecessarily admitting and treating those patients with negative fibronectin results. 

However, the study showed that the hospital stay was shorter for patients with negative results than those with positive results. We believe that having a negative FFN result will also support early discharge from the hospital. The average hospital length of stay was 3 and 4 days, respectively. This one-day difference cost the hospital around USD 10,000 extra.

We also noticed the significant difference in intervention (medications) in the two groups of women admitted with threatened preterm labor, which was significantly higher in the FFN positive test group. We believe that this is mainly due to the unconscious effect of the FFN positive values on the physician’s preference toward unnecessary intervention. In other words, steroids or tocolytics tend to be administered to women directly after admission before a re-assessment if FFN was positive; on the other hand, the proper re-assessment of women admitted with threatened preterm labor was conducted before the administration of any medication if FFN was negative. 

Surprisingly, 23 (45%) women admitted with preterm labor developed high blood sugar secondary to corticosteroid administration, which was unnecessary as they delivered at term eventually. Rising blood sugar in pregnancy even if temporary can have potential side effects on the mother or her fetus, besides the significant disturbance to the mother due to multiple pricking for sugar monitoring. 

A comparative cost analysis was conducted from March to August 2015 to the same period in the following year (after implementing FFN) to estimate the cost effectiveness of the test in reducing the overall hospital cost. The cost was estimated in terms of high cost and low cost. The low estimated cost is calculated with the inclusive cost of the FFN test and the short-stay admissions charges. The high cost is estimated by adding the room rental charges for a 3-day stay and medication charges for treating the threatened preterm labor with routine diagnostic tests.

As shown in Figure 3, there was a 14% and 17% drop in terms of low costs and high costs, respectively, due to the 7% reduction in the threatened preterm labor admissions with the introduction of an FFN test, which resulted in a significant cost saving. It is worth mentioning that preterm delivery in our hospital stayed almost the same (10% and 11.5%) before and after the test, respectively, which corresponds to the CDC reports from 2008 [32] (12.3%). This overall cost reduction, though evident, was lower than our expectation, especially in the context of a large tertiary governmental hospital. Our understanding and analysis of this can be summarized as follows: the unspecificity of the presenting symptoms of threatened preterm labor may increase the use of unnecessary FFN test kits to support the diagnosis prematurely, which was added to the cost [33]; the absence of risk stratification tools or checklists, so women can be categorized as being at low- or high-risk for preterm delivery groups which will decrease the need for using FFN unnecessarily for the high-risk group; and finally, the unconscious effect on a physician’s decision where positive FFN acted as a motive to admit or take extra precautions such as medication without clinical indication. This pointed to the possible limitations of our study: cervical length measurement was not considered in the initial assessment which could have allowed us to compare two categories regarding the cost-effectiveness. We included the direct year after FFN implementation where the new protocol compliance may still not be that strong, and no grouping of women into low- and high-risk for preterm delivery with individual cost analyses.

## 6. Conclusions

This was the first study to analyze the cost of FFN implementation to triage women with threatened preterm labor on reducing hospital admissions in a tertiary hospital in the United Arab Emirates (UAE). Its usage was simple, fast and does not require extra skills with high negative predictive values. However, the cost reduction was evident but not that large. We feel that applying such a test should be more cost-effective if applied according to women’s risk tendency to preterm delivery. However, more long-term follow up studies taking into consideration the effect on the physician’s decision, cost-effectiveness compared to cheaper methods alone such as cervical length and its role in different risky groups are needed before adapting it universally as a routine cost-effective test for reducing hospital admissions in women with preterm labor. 

## Figures and Tables

**Figure 1 jpm-13-00894-f001:**
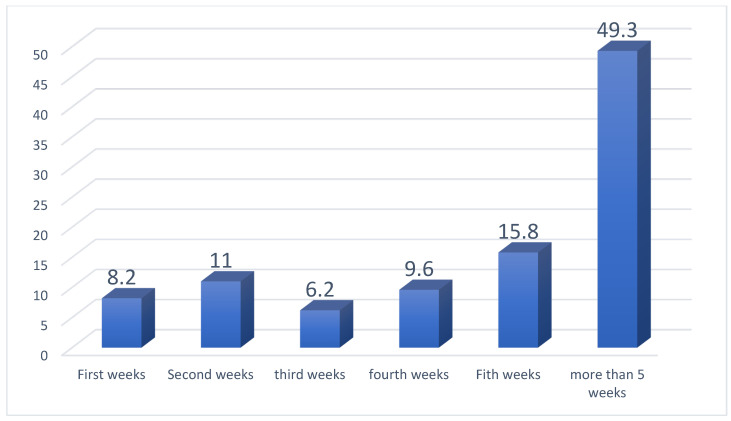
Women who were FFN negative and delivered preterm. Weeks are the time from FFN test collection date until delivery date.

**Figure 2 jpm-13-00894-f002:**
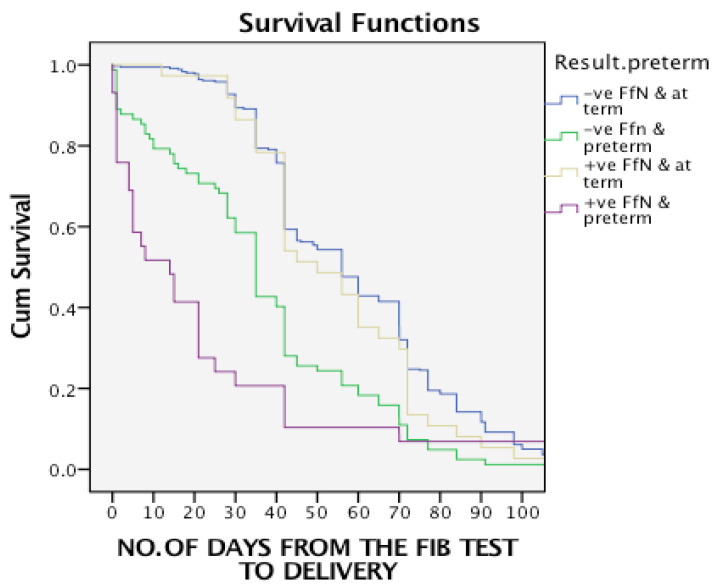
Survival curve for the four sub-groups. Log rank (Mantel–Cox, chi-square = 167.232 and *p*-value < 0.001).

**Figure 3 jpm-13-00894-f003:**
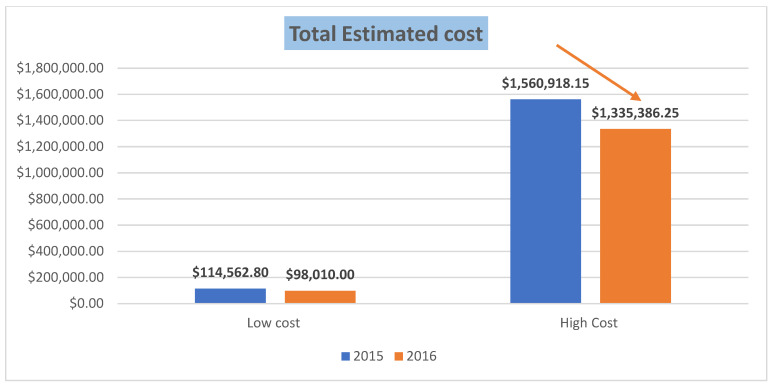
Comparative cost analysis.

**Table 1 jpm-13-00894-t001:** Sensitivity, specificity, positive predictive value (PPV) and negative predictive value (NPP) of fetal fibronectin test for preterm delivery.

Fetal Fibronectin Test	Preterm Delivery	Delivery at Term	Total
Negative	155	586	741
Positive	42	57	99
Total	197	643	840

Exact Fischer test, chi (1)-square = 10.399, *p*-value is 0.001.

**Table 2 jpm-13-00894-t002:** Comparison of delivery conditions according to fetal fibronectin result median [Q1–Q3].

	Positive for Fetal Fibronectin		Negative for Fetal Fibronectin		*p*-Value
	At-Term	Preterm	At-Term	Preterm	
	(*n* = 57)	(*n* = 42)	(*n* = 586)	(*n* = 155)	
Gestational age at fetal fibronectin test	31 (27.2–33)	31 (29–33)	31 (28–32.5)	30 (28–32)	0.223
Gestational age at delivery	39 (38–39)	34 (31–36)	39 (38–40)	36 (35–37)	<0.001
Days between sampling and delivery	42 (38.5–72)	18 (1.5–42)	60 (42–77)	42 (28–63.75)	<0.001
No. of parity	2 (0–3)	1 (0–3)	1 (0–2)	2 (0–3)	0.001
No. of days in hospital *	3 (2–4)	4 (2–8)	2 (1–3)	3 (2–5)	0.004

* Women admitted to hospital with threatened preterm labor until discharge without delivery. Kruskal–Wallis test, any *p*-value < 0.05 considered as statistically significant between groups.

**Table 3 jpm-13-00894-t003:** Comparison of management provided by fetal fibronectin test and preterm delivery.

Fetal Fibronectin Test	Variables	Categories	At-Term	Preterm	*p*-Value
Negative	Action taken after the result	Discharge	448 (77)	104 (67.5)	0.012
		Admitted	134(23)	50 (32.5)	
	Steroid medication given	No	272 (89.2)	81 (77.9)	0.004
		Yes	33 (10.8)	23 (22.1)	
	Atosiban medication given	No	298 (97.7)	100 (96.2)	0.297
		Yes	7 (2.3)	4 (3.8)	
	High blood sugar after steroid	Normal	19(61.3)	11 (55)	0.437
		Impaired	12 (38.7)	9 (45)	
	Insulin medication given	No	20 (62.5)	15 (68.2)	0.447
		Yes	12 (37.5)	7 (31.8)	
Positive	Action taken after the result	Discharge	16 (28.1)	9 (21.4)	0.304
		Admitted	41 (71.9)	33 (78.6)	
	Steroid medication given	No	19 (44.2)	11 (29.7)	0.136
		Yes	24 (55.8)	26 (70.3)	
	Atosiban medication given	No	41 (95.3)	29 (78.4)	0.025
		Yes	2 (4.7)	8 (21.6)	
	High blood sugar after steroid	Normal	10 (45.5)	16 (66.7)	0.125
		Impaired	12 (54.5)	8 (33.3)	
	Insulin medication given	No	11 (50)	16 (66.7)	0.199
		yes	11 (50)	8 (33.3)	

Exact Fischer chi-square test, for testing the association between variables.

**Table 4 jpm-13-00894-t004:** Cost of hospital stay and any possible interventions for any patient admitted with suspected preterm labor.

Cost Elements	Average Cost (in USD)	Total Cost (in USD)
Room cost		
Room stay (single room)/day Length of stay (3 days)	136.13	408.39
Routine lab Investigations		
CBC	13.61	175.6
Blood group and save	35.39	
MSU and culture	27.23	
HIV	35.39	
VDRL	13.61	
HBsAg	23.14	
HVS	27.23	
Total		
Other diagnostics		
CTG (Performed twice) Serial blood sugar Obstetric ultrasound (once) Total	108.90 20.42 68.06	197.38
Medications		
Atosiban (complete course, 5 doses)	860.34	904.44
Betamethasone (complete course, 2 doses)	29.40	
Insulin Lispro 300 units cartridges (with 0.5 units insulin delivery device)	11.98	
IV fluids		
Total	2.72	
Total Cost	$1686.41	

## Data Availability

The data presented in this study are available in tabulated form on request. The data are not publicly available due to ethical restrictions and project regulations.

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
