# Peer review of "Using Fetal Fibronectin Test to Reduce Hospital Admissions with Diagnosis of Preterm Labor: An Economic Evaluation Study"

_jpm, 2023, doi:10.3390/jpm13060894_

Round 1

Reviewer 1 Report

This is cohort study to validate the usefulness of Effect of Fetal Fibronectin on Reducing Health Resources for Women with pre term Labour in a Ter-tiary Hospital in UAE

The positive results of the presented work are probably real, but the path followed to obtain them suffers from defects:

- Despite the calculation of the sample was not performed, the sample size insufficient to determine normal values of parameters. Also, the cases should be stratified according to gestational age.

- In the statistical methodology it is not determined if parametric studies should be carried out or not, and it is decided to show the results with both methodologies, in a mixed version. In addition, clearly quantitative variables are categorized, thus losing sensitivity in the statistical treatment.

- In the Discussion, the narrative thinking to obtain the conclusion is not clear.

- In the Bibliography, references are very old.

The idea discussed a lot in the lecturer even since 2008, what is the new results you added

Cochrane Database Syst Rev. 2008 Oct; 2008(4): CD006843. Published online 2008 Oct 8. doi: 10.1002/14651858.CD006843.pub2

PMCID: PMC6492504PMID: 18843732

Fetal fibronectin testing for reducing the risk of preterm birth

Vincenzo Berghella,corresponding author Edward Hayes, John Visintine, and Jason K Baxter

Author Response

Response to Reviewer 1 Comments

Point 1: - Despite the calculation of the sample was not performed, the sample size insufficient to determine normal values of parameters. Also, the cases should be stratified according to gestational age.

Response 1: study was conducted on the period (September 2015-December 2016), all the cases eligible to the inclusion criteria (singleton with 24-34 weeks of gestation) were included at that period, and this is making our sample 840 women. We documented all the gestational age but we didn’t stratify them as we didn’t feel that this will affect the outcome of the study (the cost-effect). However we showed how the FFN test predictive value is changing according to the weeks passed from the date of the test to delivery date .

Point 2: In the statistical methodology it is not determined if parametric studies should be carried out or not, and it is decided to show the results with both methodologies, in a mixed version. In addition, clearly quantitative variables are categorized, thus losing sensitivity in the statistical treatment.

Response 2: Thank you for your question, we added a section for statistical analysis, and we mentioned why nonparametric statistical methods were used in the analysis.

Point 3: In the Discussion, the narrative thinking to obtain the conclusion is not clear

Response 3: we re-wrote this part

Point 4: In the Bibliography, references are very old.The idea discussed a lot in the lecturer even since 2008, what is the new results you added

Response 4: we updated the references in concordance with our study.

 What we feel we added..

- this is the first cost-analysis study to be done in UAE and the MENA region according to our knowledge.

- Implementation of a local protocol or guidance should be assessed well in term of efficacy and cost effectiveness especially in tertiary and governmental hospitals. FFN testing didn’t show that high reduction in cost as it would’ve been expected. sometimes simple and cost-effective measures can be sufficient. we feel that FFN test should be reserved for the women with low risk to preterm delivery rather than high-risk women. A note to be taken in consideration for the long term follow up studies

Reviewer 2 Report

It is a study highlighting the importance of fibronectin testing as a screening test for admission for symptomatic patients between 24 and 34 weeks of gestation in order to decrease the healthcare costs for hospital admission. The authors have analysed 840 cases and demonstrated a 7% reduction of cases with clinical symptomatology of preterm labor as described, especially pain  : uterine contraction, low back pain, pelvic pressure, or low abdominal pressure).

The title is not appropriate: I suggest to add Fetal Fibronectin testing and women with preterm labour symptomatology because if the test is negative and no clinical manifestation , it is not a preterm labor.

The abstract must reflect the content of the article. I suggest to revise.

The big problem of this manuscript is the methodology of study . The inclusion criteria are wrong , because , in 2023 didn’t include cervical length measurement as a primary screening for Fibronectin testing. Also, the references are old.

I suppose Tractocele is Tractocile, Atosiban, Rectocele is Tractocele, etc please revise the text.

Lines 140-141- I don’t understand the length of hospital stay, I suppose the stay since the delivery time.

I suggest to present the local protocol for Threatening Premature birth.

Corticotherapy is very controversial also.

Author Response

Response to Reviewer 2 Comments

Point 1: It is a study highlighting the importance of fibronectin testing as a screening test for admission for symptomatic patients between 24 and 34 weeks of gestation in order to decrease the healthcare costs for hospital admission. The authors have analysed 840 cases and demonstrated a 7% reduction of cases with clinical symptomatology of preterm labor as described, especially pain  : uterine contraction, low back pain, pelvic pressure, or low abdominal pressure).

The title is not appropriate: I suggest to add Fetal Fibronectin testing and women with preterm labour symptomatology because if the test is negative and no clinical manifestation , it is not a preterm labor.

The abstract must reflect the content of the article. I suggest to revise.

Response 1: Thanks for your comments; as we wanted to see the effect of implementing the FFN test at our local protocol on reducing the hospital admissions, so we included all women with possible symptoms of pre term labor to do the test to see if we can use is as a triage for those women. Women with no such symptoms, or\with no clinical manifestation were not included for the test.

We adjusted the title to better reflect the article content to (Using Fetal Fibronectin Test to reduce hospital  admissions with Diagnosis of Preterm Labor, an economic evaluation study.)

abstract also adjusted 

Point 2: The big problem of this manuscript is the methodology of study . The inclusion criteria are wrong , because , in 2023 didn’t include cervical length measurement as a primary screening for Fibronectin testing. Also, the references are old

Response 2:

- we rewrote the methodology.

 - As this study included the period of (2015-2016), where the FFN testing first implemented in our local protocol. the test was done for all women presented to emergency during that period with possible symptoms or signs of pre term labor . cervical length assessment was not considered a must to be done though cervical assessment for dilatation was done. however, in our limitation we documented that the cervical length measurement was not included. this mostly will be the topic of coming studies were we can compare the cost effectiveness of both parameters alone or in conjunction together.

- References were updated in concordance with our study

Point 3: I suppose Tractocele is Tractocile, Atosiban, Rectocele is Tractocele, etc please revise the text.

Response 3: Thaks, All revised ; Atosiban

Point 4: Lines 140-141- I don’t understand the length of hospital stay, I suppose the stay since the delivery time.

Response 4: Added the explanation in the text ; Length of hospital stay ( if patient admitted ; then the length of the stay is the days she spent at the hospital form date of the FFN test was done till discharge from the hospital without being delivered )

Point 5: I suggest to present the local protocol for Threatening Premature birth

Response 5:Thanks for the suggestions ; local protocol added at last part of the introduction

Point 6: Corticotherapy is very controversial also.

Response 6:  we didn’t discuss the efficacy of different part of the treatment as part of the local protocol , we analyze the cost  resulted from its use especially if lead to unnecessary elevated blood sugar and administration of insulin

Round 2

Reviewer 1 Report

Thank you for your revision

Author Response

thanks for your comments and review 

Reviewer 2 Report

no new comments 

It is important for a study to be reliable and to be generalised. 

Author Response

Thanks for your comments 

we updated the manuscript with English review